

# OES-Fed: a federated learning framework in vehicular network based on noise data filtering

Yuan Lei[1], Shir Li Wang[1], Caiyu Su[2] and Theam Foo Ng[3]

[1] Faculty of Art, Computing and Creative Industry, Universiti Pendidikan Sultan Idris, Tanjong Malim, Perak, Malaysia
[2] Guangxi Vocational & Technical Institute of Industry, Nanning, Guangxi, China
[3] Centre for Global Sustainability Studies, Universiti Sains Malaysia, Penang, Malaysia

## ABSTRACT

The Internet of Vehicles (IoV) is an interactive network providing intelligent traffic management, intelligent dynamic information service, and intelligent vehicle control to running vehicles. One of the main problems in the IoV is the reluctance of vehicles to share local data resulting in the cloud server not being able to acquire a sufficient amount of data to build accurate machine learning (ML) models. In addition, communication efficiency and ML model accuracy in the IoV are affected by noise data caused by violent shaking and obscuration of in-vehicle cameras. Therefore we propose a new Outlier Detection and Exponential Smoothing federated learning (OES-Fed) framework to overcome these problems. More specifically, we filter the noise data of the local ML model in the IoV from the current perspective and historical perspective. The noise data filtering is implemented by combining data outlier, K-means, Kalman filter and exponential smoothing algorithms. The experimental results of the three datasets show that the OES-Fed framework proposed in this article achieved higher accuracy, lower loss, and better area under the curve (AUC). The OES-Fed framework we propose can better filter noise data, providing an important domain reference for starting field of federated learning in the IoV.

# INTRODUCTION

*The rotten apple injures its neighbors.——An old proverb*

The Internet of Vehicles (IoV) is an interactive network consisting of an inter-vehicle network, intra-vehicle network, and vehicular mobile internet data (*Raza et al., 2021*; *Gunagwera & Zengin, 2022*). The IoV collects information by using wireless communication devices such as GPS, RFID, sensors, cameras other in-vehicle devices, then transmits the vehicle's local data to the server. The server receives these local data, then uses related technology to analyse and process it, and finally provides intelligent traffic management, intelligent dynamic information service, and intelligent vehicle control to the running vehicles.

Corresponding authors
Shir Li Wang,
shirli_wang@fskik.upsi.edu.my
Caiyu Su, suyu2210@yeah.net

Nowadays, most IoV models are established using cloud-based servers, which facilitates the analysis of the collected vehicle image data and the results back to the running vehicles. In the near future, a vehicle with sensors and cameras will collect at least 10 terabytes of daily data. According to the statistics of the global market of the IoV, it can be seen that it reached 245.42 billion dollars in 2015, growing to 643.44 billion dollars in 2020, and is expected to exceed 1.5 trillion dollars in 2025. Meanwhile, the global IoV penetration rate increased from 30.7% in 2018 to 45% in 2020, and is expected to cover approximately 60% of vehicles globally by 2025. As a result, the data generated by vehicles will also increase dramatically, while a myriad of local vehicle data is a heavy load for the IoV. For the transmission efficiency of a wireless networks, many researchers have proposed their own solutions (*Chithaluru, Tiwari & Kumar, 2021*; *Chithaluru et al., 2021b*). For example, *Chithaluru et al. (2021a)* proposed a process which includes a collection of physical environmental parameters on a single board computer-based to help improve the efficiency of a wireless network. *Singh et al. (2022)* proposed a Hybrid Genetic Firefly Algorithm-based Routing Protocol to solve the optimal routing of sparse, dense and real traffic network scenarios in vehicular *ad-hoc* network (VANET). In addition, the data is vulnerable to issues such as unstable networks and vehicle high speed during transmission, so the local data of vehicles used for training is redundant for the IoV. *Wang, Song & Liu (2020)* argued that 15–20% of the data in the vehicle network is noise data, while *Ghane et al. (2020)* believed that the percentage of the noise data should be 18–25%. The isolated data island has arisen due to data privacy issues that have received high attention (*Yang et al., 2021a*, *2021b*). The isolated data island can lead to less data being collected, yet machine learning models need more data for training in order to achieve excellent accuracy (*Burlet & Hindle, 2017*; *Baumer, 2018*; *Wang et al., 2021*).

Based on the above problems, the use of federated learning emerges as a feasible solution. Federated learning is a distributed machine learning paradigm proposed by Google (*Tehseen, Farooq & Abid, 2021*; *Lakhan et al., 2021*; *Novikova et al., 2022*). Instead of centralizing the dataset in a particular data centre, it chooses to acquire local machine learning (ML) model parameters trained by the client on its private local data and then aggregates the parameters only. This approach does not involve the private local data itself, only the parameters of the local ML model are transmitted. Furthermore, training the ML model locally on the client side not only decentralizes the computational tasks from the cloud service centre, but also saves communication costs and reduces the network burden. Regarding the IoV and federation learning, *Du et al. (2020)* discussed the significance and technical challenges of applying federated learning in the IoV and pointed out future research directions. For combining the IoV with federated learning, *Posner et al. (2021)* proposed a novel federated vehicle network (FVN) concept that improves the scalability and stability of IoV communication.

## LITERATURE REVIEW

### Federated learning and isolated data islands

As an emerging information and communication technology widely used in everyday transportation, the IoV generates a large amount of data and information interaction

during its operation. However, there is an increasing focus on data privacy, leading to the isolated data island in IoV. To address this problem, many scholars have incorporated some federated learning frameworks into the IoV network. *Pokhrel & Choi (2020b)* proposed a new efficient communication and privacy-preserving federated learning framework for further improving the efficiency of the IoV training data. However, the framework lacks an analysis of the loss and accuracy incurred in training the network model data. *Bao et al. (2021)* used distributed approaches to designate some vehicles as edge vehicles and used the edge vehicles as federated learning clients for local model training, resulting in an efficient deep learning network architecture. Also, *Zhao et al. (2020)* advanced federated learning and local differential privacy (LDP), as well as four LDP mechanisms to scramble the gradients generated by the vehicles, to provide high accuracy with a small privacy budget. The literature (*Pokhrel & Choi, 2020b*; *Bao et al., 2021*; *Zhao et al., 2020*) amply illustrates that the combination of vehicular networking and federated learning can be a solution to the problem of isolated data islands in the IoV. Although the models in the literature can improve the training accuracy of the network models, the global models are susceptible to noise data, leading to biases that make the global models inefficient to learn and slow to converge.

## Solutions of the noise data

To face the noise data problem, a reasonable data filter is needed to increase the training efficiency of network models, which is one of the important aspects of network model training. Data filters play an important role when dealing with large-scale datasets with a large number of samples and features (*Toet, 2016*; *Goudarzi & Rahmani, 2021*; *Li, Yang & Wen, 2021*). We found that the noise data can be filtered from both the current and the historical perspectives of the noise data.

- **Current perspective: data outlier**

Data outliers are the most used and powerful method for noise data processing. In data analysis, outliers are deviated and unexpected observations. *Reunanen, Räty & Lintonen (2020)* proposed a method to optimize anomaly detection integration using a limited number of outlier samples, which improves the efficiency of outlier detection by defining the limited outliers. However, this method requires some manual adjustment of the parameters or setting them according to the rule of thumb. Besides, *Goh, Chiew & Foo (2020)* introduced a novel method based on combined distances, capable of high accuracy outlier detection. But this did not work for outliers in high-density regions. Based on these problems, *Li, Wang & Guan, 2019* proposed a graph-based outlier detection method that can significantly improve the performance of existing outlier detection methods. The method does not distinguish between local outliers and global outliers. Therefore, a new outlier detection algorithm K-Nearest Neighbor-Local Outlier Factor (KNN-LOF) was proposed by *Xu et al. (2022)*. The K-nearest neighbour algorithm is used to region the outlier attributes, calculate the average sequence distance from data objects in the hierarchy, and redefine the reachable distance of the objects to introduce new local outlier factors. The outlier detection method is convenient and efficient and can be used for

most data, which is superior to other methods. The detection results of the outlier detection method are constrained by the clustering method itself. There is no universally applicable clustering method for different data sets, so corresponding adaptations are required for the IoV scenario and data parameters.

- **Historical perspective: data iteration**

Kalman filter and exponential smoothing are often used to filter noise data from data iteration. *Seth, Swain & Mishra (2018)* used the traditional Kalman filter to estimate the position and trajectory of a single target in motion, and obtained the actual trajectory by connecting the center of the obtained moving object image. For the processing of vehicle-related data in the IoV, *Zhang et al. (2018)* used Kalman filter for the selection of data from simple inertial navigation and data from various positioning system sources with different errors, which can effectively improve accuracy and reliability. Exponential smoothing is a way to simplify the classification process. Therefore, *Rahajoe (2019)* adopted a new feature matrix as the basis for selection and used exponential smoothing to encapsulate it with the Genetic Algorithm Support Vector Machine (GASVM) method, which significantly improved the accuracy and the number of filtered parameters. In the big data era of the IoV, multiple perspectives of analysis may yield completely different results due to the diversity of data sources. Therefore, we needed to combine the Kalman filter and exponential smoothing to ensure the accuracy of the filtering results according to the actual situation of the IoV.

Since federated learning is an emerging field, its use in handling noise data is rarely covered. So this article refers to *Ahmed et al. (2020)*, *Ye et al. (2020)*, *Li, Wang & Guan (2019)*, *Xu et al. (2022)*, *Seth, Swain & Mishra (2018)*, *Zhang et al. (2018)* for a comparative analysis of federated learning algorithms applied to different domains with the mechanism proposed in this article, as shown in Table 1.

## SCENE DESCRIPTION

Figure 1 shows a real-life Vehicle to Vehicle (V2V) scenario of federated learning applied in the IoV, where the vehicle transmits local ML model parameters to the central server *via* the access point provider after training local models, which solves the problems of network latency, poor model computing power and low efficiency in IoV data training. Vehicles pass the intersection smoothly in around 1–3 min. Therefore it is questionable how the approach proposed in this article can accomplish the operation in such a short time. Fortunately, manufacturers represented by Tesla, Audi, and Mercedes-Benz in the new generation vehicle have increased the intelligent driving arithmetic power to 500–1,000 Tops level. This is sufficient to support federated learning and will not result in asynchronous model training and transmission. In other words, we do not need to consider the vehicle's computing power or time. However, as shown in the top left corner of Fig. 1, there are still issues to be resolved.

**Challenge 1:** Vehicles in the IoV network are reluctant to share data, creating isolated data islands. This results in the cloud server of the IoV network not being able to acquire a sufficient amount of data to build accurate ML models to complete the calculation tasks. Even if

**Table 1 Comparison of federated learning and noise data filtering algorithms for different domains.**

|  | Technologies | Application scenario | Innovations |
|---|---|---|---|
| *Ahmed et al. (2020)* | Federated learning, Active learning | Natural disaster, Refuse classification | Modifies some federal learning model parameters and allows the machine learning (ML) model to automatically select and tag the data it learns. |
| *Ye et al. (2020)* | Federated learning | IoV | The two-dimensional contract theory is used as the distributed framework and greedy algorithm is added. |
| *Li, Wang & Guan (2019)* | Outlier detection | Data filtering | Outlier detection based on graph clustering outliers are allowed. |
| *Xu et al. (2022)* | Outlier detection, K-nearest algorithm | Data filtering | Use the K-nearest neighbor algorithm to divide different regions for outlier attributes, and then use the division of different regions for outlier attributes to introduce local outlier factors |
| *Seth, Swain & Mishra (2018)* | Kalman filter | Position and trajectory estimation of moving objects | The exponential function and the Kalman gain |
| *Zhang et al. (2018)* | Kalman filter | IoV | Kalman filter is used to fuse the position information. GPS, SINS, DR and TDOA are selected to simulate the fusion algorithm. |
| OURS | Federated learning, Outlier detection, K-nearest algorithm, Kalman filter | IoV | Outliers are detected by selecting excellent subsets, and combined with K-means algorithm, cubic exponential smoothing and Kalman filter algorithm. |

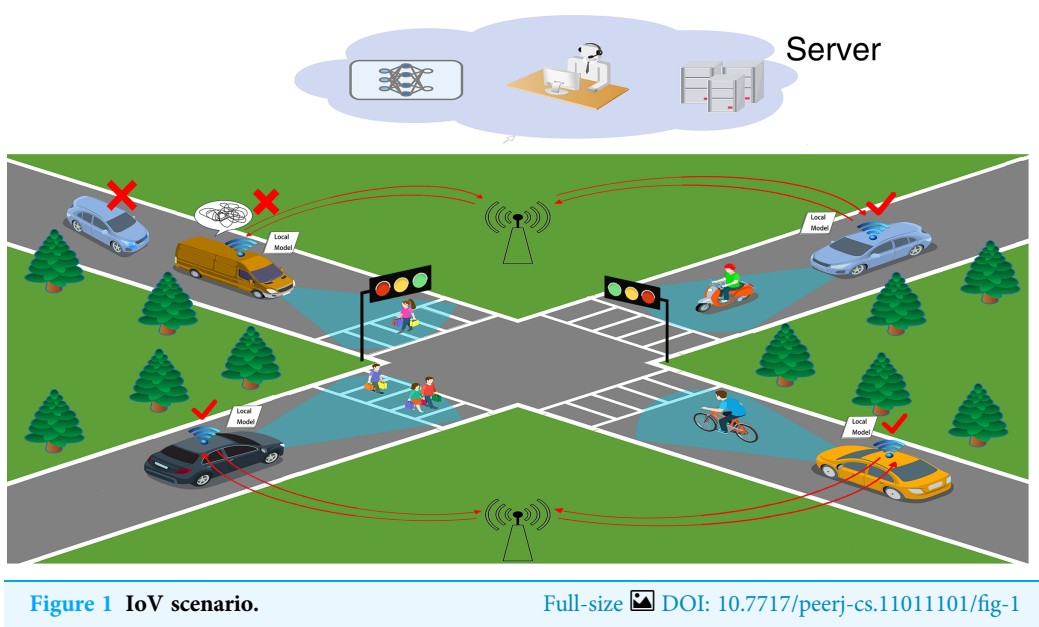

**Figure 1 IoV scenario.**               

they are willing to share data, the IoV network will still be affected by factors such as vehicle high speed and low network transmission rate, resulting in the accuracy of the ML models built by the cloud server not being able to meet the calculation tasks.

**Challenge 2:** The vehicle local ML model training effect is poor due to the problems such as violent shaking and obscuration of in-vehicle cameras. The model parameters of such clients are like "the rotten apple", which not only increases the communication

consumption, but also affects the learning efficiency and convergence speed of the global model.

**Challenge 3:** Local ML model parameters that are not relevant to the computational task have an impact on the global model. For example, the data of pedestrians on both sides of the road and shops are not relevant to the vehicle driving task. These data affect the convergence speed and accuracy of the global model.

This article defines the poorly trained ML model data and irrelevant data (in Challenges 2 and 3) as noise data in federated learning. Therefore, the key point to be explored is to alleviate the burden of the huge amount of data faced during the training of the IoV while at the same time being able to clean the noise data from the IOV data completely so as to ensure high communication efficiency and ML model accuracy of the data when combining federated learning with the IoV.

## OUR CONTRIBUTION

This article mainly focuses on the problem that the training efficiency of federated learning in the connected vehicle scenario is easily affected by noise data. We propose a new Outlier Detection and Exponential Smoothing federated learning (OES-Fed) framework.

- From the current perspective of noise data, this article provides an outlier detection method based on K-means. By screening excellent subsets and taking them as the initial clustering center, we solved the problem of significant changes in vehicle data sets in IOV, and preliminarily reduced the noise data.
- From the historical perspective of noise data, we consider the past training performance of the vehicle. Due to the influence of noise data, this article proposed the fusion of the Kalman filter and exponential smoothing to achieve the effect of noise reduction.
- After current and historical noise reduction, the actual accuracy of vehicles will be significantly affected. This article solved the problem by reintroducing the results of the iterative data filtering into the K-means algorithm and updating the filtering criteria.
- The proposed OES-Fed framework is trained on three datasets, and the global ML models are evaluated using accuracy, loss and area under the curve (AUC) metrics respectively. As a result, the OES-Fed framework proposed in this article achieved higher accuracy, lower loss, and better AUC.

## SYSTEM MODEL AND PROBLEM DESCRIPTION

### System model

This article sets up a federated learning framework composed of vehicles participating in local ML parameters sharing and the central server to simulate the real scene of safe and reliable IoV. As shown in Fig. 2, the system model includes three entities: central server, access point (AP) and vehicles. The specific functions of the different entities are as follows.

1. The central server is the core processing node of IoV, with high-speed computing capability and good scalability, which can run reliably for a long time and can undertake

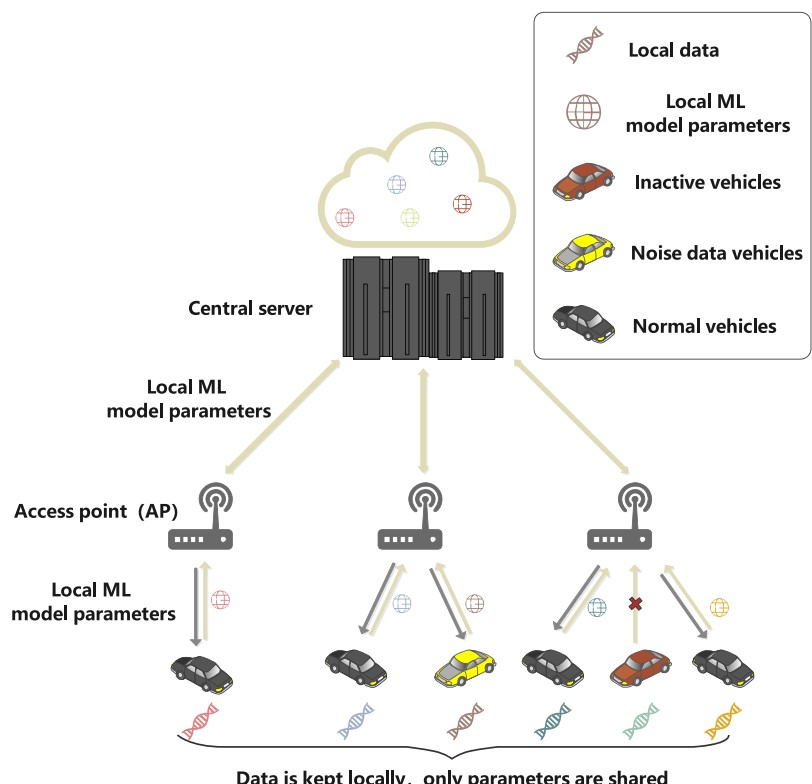

Figure 2 System model.

any level of computing work in the network. Meanwhile, the server does not receive the original data collected by the vehicle, but only the local ML model parameters transmitted by the access point (AP), which are the parameters of the ML model formed by the vehicle through training the local dataset of the vehicle client. This article assumes that the server has infinite computing power.

2. The AP is the base station or roadside unit, which is equipped with communication and computational processing. In addition, the AP adjusts the training resources of the vehicle according to the instructions of the server.

3. The vehicle will generate a large amount of driving data and related pictures at the vehicle equipment during the driving process, and the driving data will be processed and saved locally, forming the local dataset: $\text{DataLocal}_{n,i} = \left\{ x_{n,i} \in T^i, y_{n,i} \in T^i \right\}$, $x_{n,i}$ is the input sample vector for vehicle n to participate in training, and $y_{n,i}$ is the label of the input sample vector. When the server has started some task, the vehicle will train the ML model for the base and upload the local ML model parameters to the server *via* the AP.

## Problem description
In this section, considering that multiple vehicles conduct federated learning in the Internet of Vehicles to provide the data model for the server, the instability of vehicle high speed and on-board equipment, transmission bandwidth, time delay and other factors

affect the transmission of the model in the real case, so only a fraction of compliant vehicles participate in the federated learning algorithm to share local ML model parameters to improve models' quality and efficiency. The impact of vehicle selection on the performance of the federated learning algorithm is described in the following sessions.

In the above proposed model, when the central server needs to request the vehicle dataset: $Vehicle = \{V1, V2, V3, V4, \ldots Vn\}$, the vehicle privacy data of $n \in N$ for a particular road study, the central server posts the task and the nearby edge node AP receives the task to send to the vehicle dataset: $Vehicle = \{V1, V2, V3, V4, \ldots Vn\}$, and waits for the feedback of vehicle resource information. From the feedback, the central server selects the appropriate vehicles to participate in the training model. The local dataset for each vehicle: $DataLocal_{n,i} = \left\{ X_{n,i}^i, Y_{n,i}^i \right\}$, $X_{n,i}$ is the input sample vector for vehicle n to participate in training, $Y_{n,i}$ is the label for the input sample vector. The amount of data for all participating local training is $Data = \sum_{n=1}^{N} \sum_{i=1}^{I} DataLocal_{n,i}$, $i \in I$ is the i-th sample of the input. $n \in I$ is the participating vehicles and N is the set of vehicles participating in data sharing.

After the vehicle client has trained the local model, it transmits the parameters of the local model: $ParametersLocal_{n,i} = F_{n,i}^i$ to the server *via* the AP. $i \in I$ is the i-th parameter of the input and $n \in I$ is the participating vehicle. The server receives the parameters of the local model from each vehicle client and stores them in $ParametersGlobal_{n,i} = \sum_{n=1}^{N} \sum_{i=1}^{I} ParametersLocal_{n,i}$, $i \in I$ is the i-th parameter of the input and $n \in I$ is the participating vehicle. A weight parameter $W_n$ is defined to represent the local model parameters trained by vehicle n. The goal of the whole training process is to find the parameters $X_n$ and $Y_n$ obtained by the learning algorithm with training to make the model converge to achieve prediction accuracy and minimize the loss function. $X_n$ and $Y_n$ are the set of input sample vectors and the set of input sample vector labels. A convolutional neural network (CNN) uses the original image as input, which can effectively learn the corresponding features from a large number of samples. CNN can avoid the complicated feature extraction process and directly process images. With these advantages, CNN has been widely used in image processing (*Kim & Hong, 2020*; *Wang, Ma & Wu, 2020*; *Poudyal et al., 2021*). For example, *Yang et al. (2021a, 2021b)* proposed a Multi-scale Texture Difference model (MTD-Net) and Multi-scale Self-Texture Attention Generative Network (MSTA-Net) based on improved CNN, respectively, to detect forged data. Therefore, we chose the CNN as the learning algorithm in this article. The net input of the i-th parameters mapping of the first layer $Z^{(l,i)}$ is as follows.

$$Z^{(l,i)} = \sum_{d=1}^{D} W^{(l,i,d)} \otimes X^{(l-1,d)} + b^{(l,i)} \tag{1}$$

$X^{(l-1,d)}$ is the input parameters mapping for the first layer, $X^{(l-1,d)} \in R^{M \times N \times D}$. Each output parameters mapping requires D convolution kernels as well as a bias. The predicted results $\hat{Y}_{n,i}$ are as follows.

$$\hat{Y}_{n,i} = Z_{n,i}W_n^T + b_{n,i} \tag{2}$$

Therefore, the prediction accuracy function of the federated learning model is as follows.

$$Accuracy = \frac{1}{N}\sum_{i=1}^{N} \hat{Y}_{n,i} = Y_{n,i} \tag{3}$$

$\hat{Y}_{n,i}$ is the true label if it is the predicted label trained on the test set using the trained classification model. For the loss function of the federated learning, we chose the logistic regression method to describe it, then the loss function of the local model is as follows.

$$f_n(w) = \log_2\left(1 + exp\left(-Y_{n,i}w_n^T X_{n,i}\right)\right) \tag{4}$$

The objective is the formula (2).

$$minf(w) \triangleq min\left\{\frac{1}{N}\sum_{n=1}^{N} f_n(w)\right\} \tag{5}$$

Each vehicle n updates the model in round $e w_n^e = w_n^{e-1} - l\left(w_n^{n-1}\right)$

$l$ is a predefined learning rate, and then the updated model parameters are passed through the edge nodes to the central server, which trains the e round global model parameters.

$$w_e = \frac{1}{N}\sum_{n=1}^{N}\sum_{i=1}^{I} \frac{w_n^e \text{DataLocal}_{n,i}}{DataLocal} \tag{6}$$

The probability of each vehicle being selected is as follows.

$$p_n = \sum_{i=1}^{I} \frac{\text{DataLocal}_{n,i}}{DataLocal} \tag{7}$$

We validated the model capability by calculating the AUC values of the completed global model for the training, setting the dataset with a total of M positive samples, N negative samples, and M + N predictions $\hat{Y}_{n,i}$.

$$AUC = \frac{\sum_{i=1}^{N} 1\left\{\hat{Y}_{n,i} = Y_{n,i}\right\}}{M \times N} \tag{8}$$

According to formulas (1) and (2), vehicles with highly accurate and reliable local ML model parameters can converge local loss functions $f_n(w)$ and global model parameters $f(w)$ faster. From the formula (3), it can be seen that the global ML model parameters trained by the central server depend on the local ML model parameters transmitted by the vehicle, as well as the quality of the training dataset. On the contrary, the global ML model parameters in turn determine the local model updates. Therefore, the local ML model

**Table 2 Parameter definition of OES-Fed algorithm.**

| Symbols | Definition |
| --- | --- |
| r | Global communication rounds |
| D | Discard rate |
| V | Vehicle set |
| v | Vehicle |
| m | Vehicle weights |
| $Acc_v$ | Vehicle accuracy |
| global | Global model |
| $Acc_g$ | Global model accuracy |
| step | Vehicle training resources |

**Table 3 Parameter definition of Outlier algorithm.**

| Symbols | Definition |
| --- | --- |
| e | Clustering parameters |
| D | Sample distance formula |
| d | Sample dimension |
| $\lambda$ | Arbitrary real numbers |
| k | Number of clustering centers |
| $A_j$ | Center of clustering |
| $n_j$ | Number of samples of class j |

parameters must be selected with high accuracy, low loss and better AUC to achieve convergence between local and global model updates in fewer iterations. In addition, the learning efficiency of federated learning can be improved significantly.

## OUR PROPOSED FRAMEWORK: OES-FED

As mentioned above, there is a myriad of data in the IoV. Moreover, there is a lot of noise data in the IoV due to problems such as unstable networks, signal delays or obscuration of in-vehicle cameras. The noise data also affected the modelling quality of the IoV network model, and reduced the accuracy and efficiency of image recognition. This article used federated learning as the base network model, whose expressions are shown below.

$$g_{r+1}(w) = \sum_{k=1}^{K} \frac{n_k}{n} F_{k,r}(w), \quad F_{k,r}(w) = \frac{1}{n_k} \sum_{i \in \mathscr{P}_k} c_{i,r}(w) \tag{9}$$

$g_{r+1}(w)$ is the global model parameter for round r+1, and $F_{k,r}(w)$ is the label after averaging the clients for round r. $n$ is the number of clients used for averaging, $K$ is the set of all clients, and $c_{i,r}(w)$ is the model parameter for each client in round r.

Tables 2–4 show all the notations and definitions used in this article. Figure 3 shows the overall training process of the OES-Fed framework proposed in this article.

**Table 4 Parameter definition of ESmooth algorithm.**

| Symbols | Definition |
| --- | --- |
| X | Filtered processing value |
| P | The variance value corresponding to X |
| K | Filtering gain value |
| S | Smoothing value |
| R | Smoothing period |
| $\alpha, \beta, \gamma$ | Exponential smoothing coefficient |

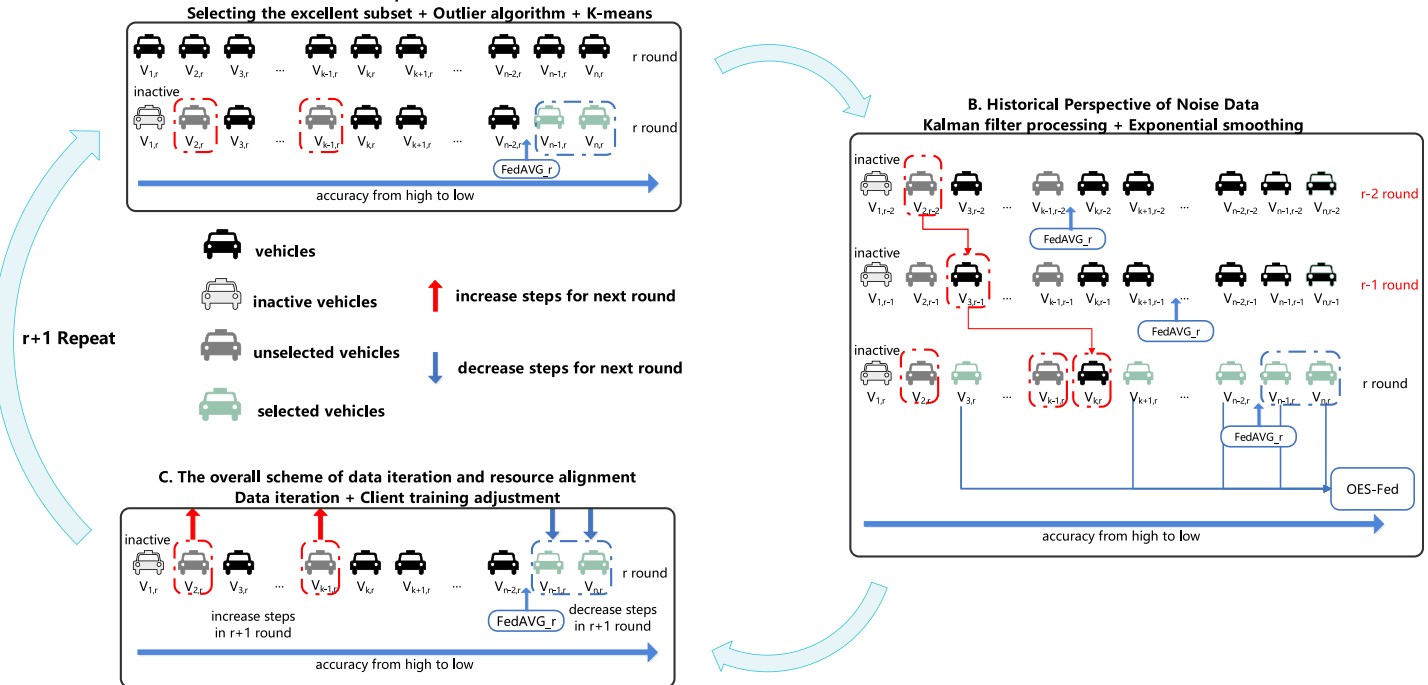

**Figure 3 OES-Fed framework.**

As shown in Fig. 3 we divided the overall training process of the initialized framework into three steps. Step A is the current perspective of nosie data with the K-means algorithm based on outliers, Step B is the historical perspective of noise data with exponential smoothing based on Kalman filtering and Step C is the overall scheme of data iteration and resource alignment. Initialization stage: the obtained total clients $(V_{1,r}, V_{2,r}, V_{3,r}, \ldots, V_{k-1,r}, V_{k,r}, V_{k+1,r}, \ldots, V_{n,r}, V_{n-1,r}, V_{n-2,r})$ are subjected to ordinary federated averaging to obtain the initial client accuracy and the initial accuracy of the global model (FedAVG).

## Phase A: current perspective of nosie data

### *Motivation: current perspective of nosie data filtering*

In this article, we chose the outlier point algorithm as the main filtering algorithm for noisy data processing. We propose a new K-means algorithm based on the combination of the

outlier algorithm and K-means algorithm. As shown in Step A in Fig. 3, $V_{1,r}$ is an inactive vehicle, so it will not enter the training round r. Finally, by training the vehicles $(V_{1,r}, V_{2,r}, V_{3,r}, ..., V_{k-1,r}, V_{k,r}, V_{k+1,r}, ..., V_{n,r}, V_{n-1,r}, V_{n-2,r})$, the outlier vehicles $V_{2,r}, V_{k-1,r}$ are selected after comparing the training accuracy of these vehicles with the accuracy of FedAVG.

### Selecting the initial clustering centre: the excellent subset

For the noise data filtering in the IoV, the original outlier detection is able to filter out some of the outliers. Still, it is but is not sufficient to better classify noise data and valid data. Therefore, in this article, based on the idea of *Li, Wang & Guan (2019)*, the performance of the outlier detection method can be improved by filtering out the good subset as the initial clustering centre of the outliers, and the algorithm expression of the method is shown below.

$$B_r = \sum_{i=1}^{N} x_i \{ Accv_{r,i} \geqslant Accg_r \} \tag{10}$$

$B_r$ is defined as the initial clustering centre of round r, $x_i$ is the model parameters of vehicle i, $Accv_{r,i}$ is the vehicle accuracy of vehicle i of round r, and $Accg_r$ is the global model accuracy of round r.

### Outlier algorithm based on excellent subset

Based on the excellent subset, this article uses the outlier algorithm to perform the initial cleaning of the noise data from the perspective of data outliers, and the algorithm expression is shown below.

$$J = \sqrt{ \sum_{j=1}^{e} \sum_{q=1}^{n_{r,j}} \sum_{a=1}^{d} \left( W_{r,j_q}^{(a)} - B_{r,j}^{(a)} \right)^2 / (n-1) } \tag{11}$$

$d$ is the sample dimension, $n_{r,j}$ is the number of samples within $j$ class in the r-round clustering result, and $e$ is the clustering parameter. $W$ is defined as the sample dataset in outlier detection, $B$ defined as the initial clustering centre, $X_i$ is the input sample vector of the sample dataset, and $Y_i$ is the label of the input sample vector.

### Current perspective on noise data: the improved K-means scheme based on the outlier algorithm

In order to achieve the deep elimination of the noise data, this article adds the K-means algorithm to the above Steps (2) and (3) to cluster the remaining clients, by calculating the distance from the outliers to each category and classifying each outlier into categories. The algorithm is able to classify the clients more effectively and accurately, thus separating the noise data from the valid data more accurately. In this article, the euclidean distance measure is used to achieve the K-means classification of the sample data. The distance formula for $W$ and $B$ is as follows.

$$D(x, y) = \sqrt{\sum_{i=1}^{d} (x_i - y_i)^2} \tag{12}$$

Therefore, the adaptation function is:

$$f(x) = \frac{\lambda}{\sum_{j=1}^{k} \sum_{x_i \in B_{r,j}} D(x_i, B_{r,j})} \tag{13}$$

$\lambda$ is an arbitrary real number, and $B_{r,j}(j = 1, 2, 3, \cdots, k)$ is the center of r-round clustering.

## Phase B: historical perspective of noise data

### Motivation: historical perspective of noise data filtering

Since the IoV data is always being updated and iterated, the client cannot be judged as good or bad from the accuracy of one training round. Therefore, both past and present data must be considered. We combine Kalman filtering with the exponential smoothing algorithm to improve the efficiency of the exponential2s smoothing algorithm. Furthermore, the exponential smoothing algorithm can better adapt to the changes in the 24 sequence itself. As shown in Step B in Fig. 3, the vehicles are reordered from the lowest to the highest accuracy, and the subscripts corresponding to the vehicles change with the number of rounds because the data are processed by Kalman filtering in each round. We eliminated $V_{k,r}$ as the outlier by comparing the actual vehicle accuracy with the r-th round accuracy.

### Kalman filter processing

This article performs a second screening of the client from the historical perspective of noise data, *i.e.*, the iterative data perspective. However, as noise data is involved, the actual performance results of the client may be biased. So this article uses the Kalman filter to perform noise reduction on the client. The exact process of the Kalman filter algorithm is shown below.

For the iterative update of the measurement data: the Kalman filter processing value obtained at moment r − 1 is set to the initial system state value at the current moment r.

$$X_{r|r-1}^{(x)} = X_{r-1|r-1}^{(x)} \tag{14}$$

$X_{r-1|r-1}^{(x)}$ $r - 1$ is the Kalman filter processing value obtained at the moment and r − 1 is the number of processing times. When $x = 1, 2, 3$, it represents the primary processing value, secondary processing value and tertiary processing value respectively. The variance value of the initial state value of the system is as follows.

$$P_{r|r-1}^{(x)} = P_{r-1|r-1}^{(x)} + Q_r \tag{15}$$

$P_{r-1|r-1}^{(x)}$ is the variance value corresponding to the Kalman filter processing values, when $x = 1, 2, 3$, it represents the variance values of the primary, secondary and tertiary processing values therein. $Q_r$ is the covariance matrix of the Kalman filter.

For each of the three Kalman filter treatments, the data is processed to obtain the Kalman filter processing value.

$$X_{r|r}^{(x)} = X_{r|r-1}^{(x)} + K_r^{(x)}\left(X_{r|r}^{(x-1)} - X_{r|r-1}^{(x)}\right) \tag{16}$$

Therefore, the three processing values of the Kalman filter are $X_{r|r}^{(1)}$, $X_{r|r}^{(2)}$, and $X_{r|r}^{(3)}$. $K_r^{(i)}$ the gain value of Kalman filter.

$$K_r^{(x)} = P_{r|r-1}^{(x)}\Big/\left(P_{r|r-1}^{(x)} + R_r\right) \tag{17}$$

Therefore, the three Kalman filter gains obtained are $K_t^{(1)}$, $K_t^{(2)}$, and $K_t^{(3)}$.

After the three Kalman filtering processes, the variance values corresponding to the Kalman filter values can be updated as follows.

$$P_{r|r}^{(x)} = \left(1 - K_r^{(x)}\right)P_{r|r-1}^{(x)} \quad (x = 1, 2, 3) \tag{18}$$

### Historical perspective of noise data: the improved exponential smoothing scheme based on Kalman filter

After the noise reduction using Kalman filtering, the exponential smoothing algorithm is used to make the results obtained by the exponential smoothing algorithm more stable. This article uses the exponential smoothing algorithm three times, combined with the client accuracy in every third round, to calculate the actual performance accuracy of the client in the third round.

For the traditional cubic exponential smoothing model in $r + R$ period:

$$\hat{S}_{r+R} = \alpha + \beta R + \gamma R^2 \tag{19}$$

$\hat{S}_{r+R}$ is the predicted value of a period, $R$ is the number of prediction periods, and $\alpha, \beta, \gamma$ are the parameter of the prediction model.

$$\left.\begin{aligned}
\alpha = &\quad 3S_r^{(1)} - 3S_r^{(2)} + S_r^{(3)} \\
\beta = &\quad a\Big[(6 - 5a)S_r^{(1)} - 2(5 - 4a)S_r^{(2)} + \\
&\quad (4 - 3a)S_r^{(3)}\Big]\Big/\left[2 - (1 - a)^2\right] \\
\gamma = &\quad a^2\left(S_r^{(1)} - 2S_r^{(2)} + S_r^{(3)}\right)\Big/\left[2(1 - a)^2\right]
\end{aligned}\right\} \tag{20}$$

Therefore, we replace the smoothing coefficients with the Kalman filter tertiary gain values in formulas (15) and (16) above. Also, use the primary processing value $X_{r|r}^{(1)}$, secondary processing value $X_{r|r}^{(2)}$ and tertiary processing value $X_{r|r}^{(3)}$ in Eqs. (15) and (16) to

substitute as the primary smoothing value $S_r^{(1)}$, secondary smoothing value $S_r^{(2)}$, and tertiary smoothing $S_r^{(3)}$, for the solution of corresponding $\alpha, \beta, \gamma$ in the formula (19).

$$\alpha = 3X_{r|r}^{(1)} - 3X_{r|r}^{(2)} + X_{r|r}^{(3)} \tag{21}$$

$$\beta = \begin{cases} \dfrac{K_r^{(3)}}{2\left[1 - K_r^{(3)}\right]^2} \left(\left[6 - 5K_r^{(3)}\right]X_{r|r}^{(1)} - \\ 2\left[5 - 4K_r^{(3)}\right]X_{r|r}^{(2)} + \left[4 - 3K_r^{(3)}\right]X_{r|r}^{(3)}\right) \end{cases} \tag{22}$$

$$\gamma = \frac{\left[K_r^{(3)}\right]^2 \left[X_{r|r}^{(2)} - 2X_{r|r}^{(2)} + X_{r|r}^{(3)}\right]}{2\left[1 - K_r^{(3)}\right]^2} \tag{23}$$

Therefore, the newly obtained $\alpha, \beta, \gamma$ are substituted into Eq. (19) to create a new cubic exponential smoothing model for the secondary screening of clients. The Improved K-means Based Outlier and Improved Kalman Filter Based Exponential smoothing are shown in Algorithm 1.

## Phase C: the overall scheme of data iteration and resource alignment
### *Motivation: the overall scheme*
As shown in Step C in Fig. 3, after the r round passes the second filtering, we take the results of the three exponential smoothing in Step B into Step A again to update the filtering criteria. We consider the clients $V_{1,r}, V_{2,r}V_{k-1,r}$ selected by the outlier optimization algorithm and the exponential smoothing optimization algorithm as the weaker vehicles by the new filtering criteria. The $V_{n-1,r}, V_{n-2,r}$ selected in Step B are considered stronger vehicles. For $V_{1,r}, V_{2,r}, V_{k-1,r}$, we increase the training resources for such vehicles to make such vehicles excellent by the increase in training resources. For $V_{n-1,r}, V_{n-2,r}$, we reduce the training resources of such clients to reduce the waste of training resources and improve training resource utilization.

### *Data iteration: (Kalman filter + Exponential smoothing) + (Excellent subset + Outlier + Improved K-means)*
When the exponential smoothing algorithm is based on Kalman filtering, the previous client-side filtering criteria are no longer valid due to changes in the actual accuracy of the client. For this reason, we introduce the results of the Kalman filter into the K-means algorithm to update the filtering criteria and accurately filter the noise data. The algorithm expression is as follows. The obtained $\hat{S}_{r+R}$ are put into the fitness function formula (12).

### *Client training adjustment*
The model combining the IoV with federated learning has computational resource optimization. Therefore, in this article, the client training resources are adjusted.
The poorer performing clients get more training resources to improve the performance of the training parameters. Meanwhile, the better performing clients reduce the load of the model by reducing epochs to improve efficiency.

**Algorithm 1: Outlier algorithm & ESmooth algorithm.**

**Outlier**$(B, W, e)$ :

  **for** *each* $a = 1,2,\dots$ **do**

$$D(x,y) = \sqrt{\sum_{i=1}^{d}(x_i - y_i)^2}$$

$$f(x) = \frac{\lambda}{\sum_{j=1}^{k}\sum_{x_i \in W_j} D(x_i, W_j)}$$

$$J = \sqrt{\sum_{j=1}^{e}\sum_{q=1}^{n_j}\sum_{a=1}^{d}\left(W_{j_q}^{(a)} - W_j^{(a)}\right)^2 / (n-1)}$$

  **end**

  *return J*

**ESmooth**$(X, x \in X)$:

  for *each round* $r = 4,5,\dots$ **do**

$$X_{r|r-1}^{(x)} \leftarrow X_{r-1|r-1}^{(x)}$$

$$P_{r|r-1}^{(x)} \leftarrow P_{r-1|r-1}^{(x)} + Q_r$$

$$X_{r|r}^{(x)} \leftarrow X_{r|r-1}^{(x)} + K_r^{(x)}\left(X_{r|r}^{(x-1)} - X_{r|r-1}^{(x)}\right)$$

$$K_r^{(x)} \leftarrow P_{r|r-1}^{(x)} / \left(P_{r|r-1}^{(x)} + R_r\right)$$

$$P_{r|r}^{(x)} \leftarrow \left(1 - K_r^{(x)}\right)P_{r|r-1}^{(x)} \quad (x = 1, 2, 3)$$

$$\hat{S}_{r+R} \leftarrow \alpha + \beta R + \gamma R^2$$

$$\alpha \leftarrow 3X_{r|r}^{(1)} - 3X_{r|r}^{(2)} + X_{r|r}^{(3)}$$

$$\beta \leftarrow \left\{ \frac{K_r^{(3)}}{2\left[1 - K_r^{(3)}\right]^2}\left\{\left[6 - 5K_r^{(3)}\right]X_{r|r}^{(1)} - 2\left[5 - 4K_r^{(3)}\right]X_{r|r}^{(2)} + \left[4 - 3K_r^{(3)}\right]X_{r|r}^{(3)}\right\}\right.$$

$$\gamma \leftarrow \frac{\left[K_r^{(3)}\right]^2\left[X_{r|r}^{(2)} - 2X_{r|r}^{(2)} + X_{r|r}^{(3)}\right]}{2\left[1 - K_r^{(3)}\right]^2}$$

  **end**

  *return $\hat{S}_{r+R}$*

$$epochs(r, i, Acc) = \begin{cases} epoch - step & \text{if } Acc_{r,v} > Acc_{r,g} \\ epoch + step & \text{if } Acc_{r,v} < Acc_{r,g} \\ & \text{and } len(epoch + step) \\ & = len(epoch - step) \\ epoch & \text{otherwise} \end{cases} \quad (24)$$

$$\int (\hat{S}_{r+R}) = \frac{\lambda}{\sum_{j=1}^{k}\sum (\hat{S}_{r+R})_i \in W_{r,j} D\left((\hat{S}_{r+R})_{i'}, B_{r,j}\right)} \quad (25)$$

The epochs required for vehicle in i-rounds are determined by $Acc_{r,v}$ in $r - 1$ rounds, and $Acc_{r,g}$ is the accuracy of the r-round global variables. The overall flow of our OES-Fed algorithm is shown in Algorithm 2.

---

**Algorithm 2: OES-Fed algorithm.**

**for** *each round r = 1,2, … * **do**

  $ms \leftarrow max(D * V, 1)$

  $SV \leftarrow (random\ set\ of\ ms\ clients)$

  **for** *each client $v \in SV$ in parallel* **do**

    $m_{r'}^v \leftarrow ClientUpdate(v, m_r)$

    $Acc_{r,v}^v \leftarrow test(m_{r'}^v)$

  **end**

  $m_{r'} \leftarrow \sum_{v=1}^{V} \dfrac{n_v}{n} m_{r'}^v$

  $Acc_{r,v} \leftarrow test(m_{r'})$

  $Acc_{r,g} \leftarrow test(global_r)$

  **if** $Acc_{r,v}^v > Acc_{r,g}$ **then**

    $B \leftarrow Acc_{r,v}$

    **else** $W \leftarrow Acc_{r,v}$;

  **end**

  **for** *each $b \in W$* **do**

    **if** $b == Acc_{r,g}$ **then**

      $Nor \leftarrow b$

      **else** $L \leftarrow b$;

    **end**

  **end**

  $out_r \leftarrow Outlier(L, B)$

  $G \leftarrow out_r$

  $X \leftarrow (remove\ out_r\ from\ L)$

  $ses_r \leftarrow ESmooth(Acc_{x,v}, w)$

  $G \leftarrow ses_r$

  $L \leftarrow (remove\ ses_r\ from\ X)$

  **for** *each client $\in V$* **do**

    **if** *client $\in L$* **then**

      $v \leftarrow +step$

      **else if** *client $\in G$* **then**

        $v \leftarrow -step$

        **else**

            *unchange step*

            *(client $\in Nor$)*

        **end**

      **end**

    **end**

  **end**

  $m_{r''}^v \leftarrow (select\ m_{r'}^v\ from\ L)$

  $m_{r''} \leftarrow \sum_{v=1}^{L} \dfrac{n_v}{n} m_{r'}^v$

**end**

*return $m_{r''}$ to server*

---

**Table 5 Parameter setting of the experiment.**

| Symbols | MNIST CIFAR-10 | Vehicle classification |
|---|---|---|
| Number of training set images | 60,000 | 2,000 |
| Number of images in the test set | 10,000 | 200 |
| Client | 40 | 40 |
| Total number of rounds | 30 | 30 |
| Number of local training rounds | 10 | 10 |
| Learning efficiency | 0.01 | 0.01 |
| Data type | non-iid | non-iid |
| Local data batch size | 64 | 64 |
| Convolution kernel | 5 * 5 | 5 * 5 |

## RESULTS

MNIST datasets are widely used in Internet of Vehicles research. For example, *Manias & Shami (2021)* used MNIST to validate their intelligent transportation systems combined with federated learning; *Yang et al. (2022)* used MNIST to validate a differentially private federated learning *via* reconfigurable intelligent surface; and *Pokhrel & Choi (2020a)* validated an autonomous blockchain-based joint learning (BFL) framework using MNIST. The CIFAR-10 dataset has numerous applications in vehicle Internet research, too. For example, *Yang et al. (2020)* used CIFAR-10 to test a federated learning framework based on air computing communication efficiency. *Husnoo & Anwar (2021)* tested the pixel attack in modern deep neural networks (DNNS) in autonomous vehicles using MNIST and CIFAR-10. Based on previous studies, we used three datasets, MNIST (*LeCun, Cortes & Burges, 2010*), CIFAR-10 (*Krizhevsky, Nair & Hinton, 2014*) and the vehicle classification dataset (*PaddlePaddle, 2021*), to verify the experiment.

This experiment used the windows platform, Intel Core i7-11700K 3.60 GHz processor, and the software anaconda 3 + Jupyter notebook + pytorch 1.4.0 + python 3.8.5.

This article conducted pre-experiments using FedAVG, FedSGD and OES-Fed as network models. For each network model setup, we used 40 clients with 10 local communication rounds and 30 global communication rounds to train on the MNIST dataset, the CIFAR-10 dataset and the vehicle classification dataset. The MNIST dataset consists of 60,000 training images and 10,000 28 × 28 pixel test images and the CIFAR-10 dataset consists of 60,000 training images and 10,000 32 × 32 pixel test images. The vehicle classification dataset is related to vehicle classification, which includes 2,000 training images and 200 test images. More specifically, the training images are divided into nine types of vehicles: bus, taxi, truck, family sedan, minibus, jeep, SUV, heavy truck, racing car and fire engine. We also used the non-iid for the dataset to better simulate the uneven distribution of the IoV data, as shown in Table 5. The loss function was defined as follows, where $P_{False}$ is the number of incorrectly parsed images for this model and $P_{True}$ is the number of correctly parsed images.

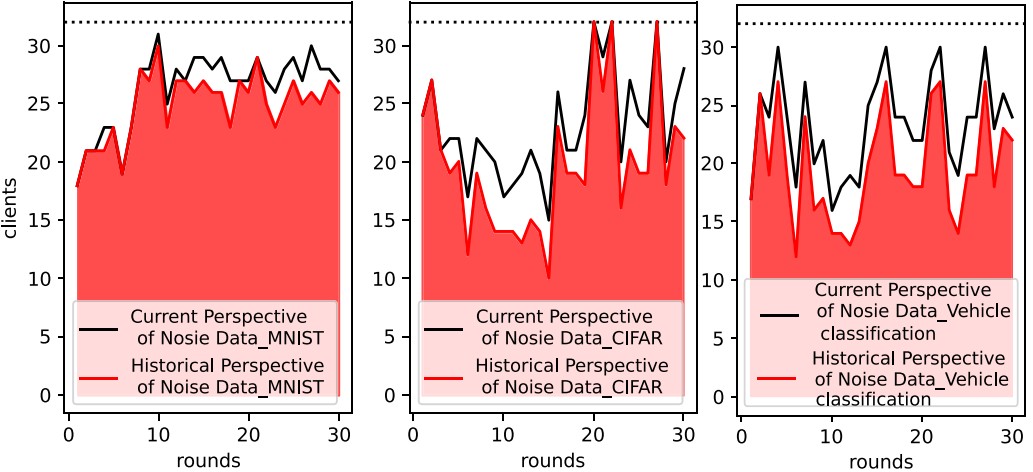

**Figure 4** Statistics on the number of actual clients in the OES-Fed model using the MNIST dataset, the CIFAR-10 dataset and the vehicle classification dataset.

$$loss = \frac{P_{False}}{P_{True} + P_{False}} \tag{26}$$

## Current and historical perspective's noise data filtering results

Figure 4 shows the actual number of clients integrated by the OES-Fed model per round of cloud server after filtering by outliers with the Kalman filter when running the MNIST dataset, the CIFAR-10 dataset and the vehicle classification dataset. In other words, the results of client filtering by current noise perspective and historical noise perspective are shown. The black line in Fig. 4 shows the number of clients per round after removing outliers from the current noise perspective, and the red part shows the actual number of clients per round in the end from the combined historical noise perspective. When using the MNIST dataset, the OES-Fed model retains as many client model parameters as possible, while the number of actual clients tends to level off as the number of rounds increases. When using the CIFAR-10 dataset and the vehicle classification dataset, more clients are screened out due to their current round performance *vs* their historical performance. As a result, outliers and the Kalman filter play a key role in the filtering of clients.

## OES-FED framework's accuracy, loss and AUC results

As shown in Fig. 5, we compared the accuracy of the three models, FedAVG, FedSGD and OES-Fed, using the MNIST and CIFAR-10 datasets. The accuracy of the three models continued to improve as the number of global communication rounds increased. For the MNIST, the model accuracy of OES-Fed improved by about 2.36% over that of FedAVG and 44.46% over that of FedSGD. For the CIFAR-10, the model accuracy of OES-Fed was approximately 42.02% higher than that of FedAVG and about 47.6% higher than that

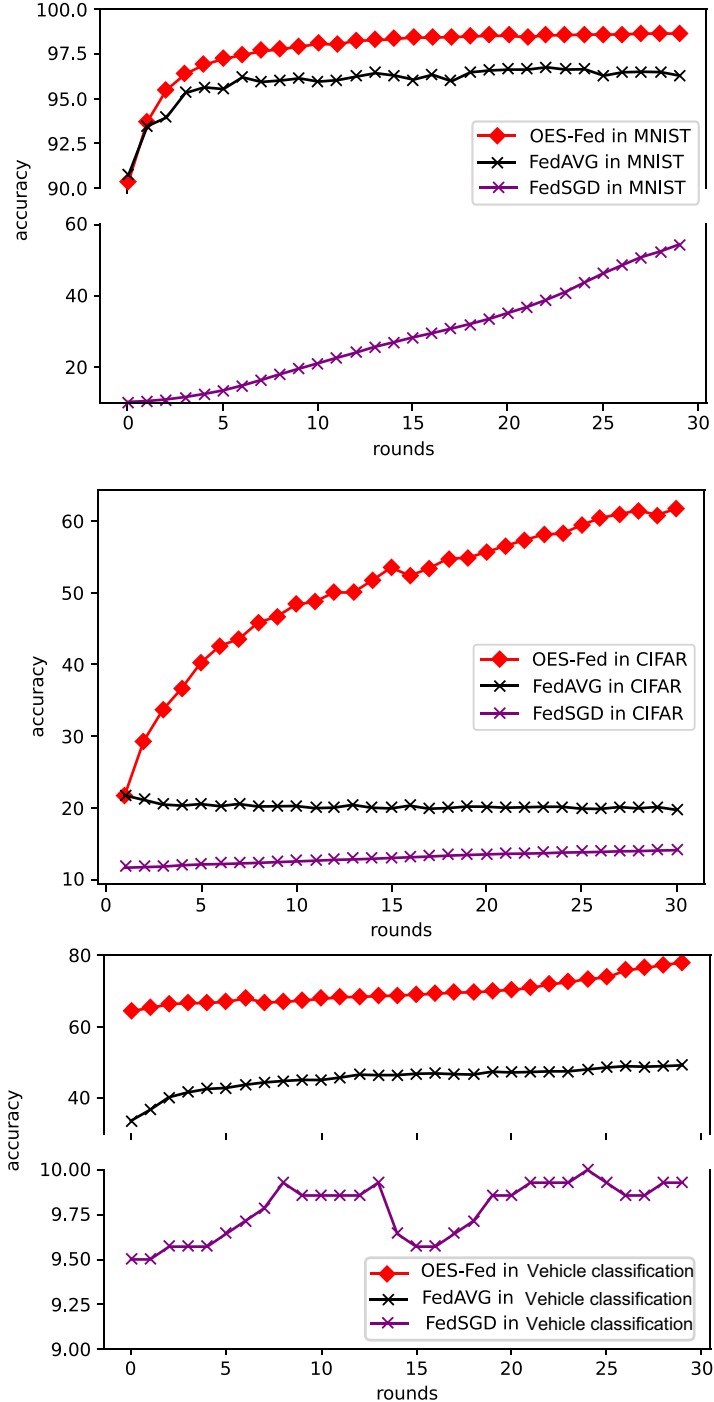

**Figure 5 Comparison of the accuracy of each model for MNIST, CIFAR-10 and the vehicle classification datasets using different models and non-iid data settings.**

of FedSGD. For the vehicle classification dataset, the accuracy of the OES-Fed was 32.4 % higher than FedAVG, and 68.0 % higher than FedSGD.

As shown in Fig. 6, we compared the loss values of the three models, FedAVG, FedSGD and OES-Fed, using the MNIST and CIFAR-10 datasets. The losses of the three models kept increasing as the number of global communication rounds increased. In the case of MNIST, the model loss of OES-Fed was approximately 0.23 lower than that of FedAVG and approximately 2.11 lower than that of FedSGD. In the case of CIFAR-10, the model loss of OES-Fed was approximately 0.38 lower than that of FedAVG and approximately 0.44 lower than that of FedSGD. For the vehicle classification dataset, the loss of OES-Fed is 0.59 lower than FedAVG and 1.27 lower than FedSGD.

As shown in Figs. 7–9, we compared the AUC values of the FedAVG, FedSGD and OES-Fed models using the MNIST, CIFAR-10 and the vehicle classification datasets. From the results shown in the figure, the AUC values of OES-Fed were 0.22 higher than those of FedAVG and 0.32 higher than those of FedSGD in the MNIST. The AUC values of OES-Fed were 0.14 higher than those of FedAVG and 0.17 higher than those of FedSGD in the CIFAR-10. In the vehicle classification dataset, the AUC value of OES-Fed was 0.27 higher than FedAVG and 0.33 higher than FedSGD.

Figure 10 shows the accuracy comparison of all clients at the final round using the OES-Fed model and the FedAVG model for the MNIST dataset and the CIFAR-10 dataset, respectively. The red line is the OES-Fed accuracy for the 40 clients in the final round, sorted from smallest to largest; the black line is the FedAVG accuracy for the 40 clients in the final round, also sorted from smallest to largest. The grey-shaded force between the red and black lines is the difference in accuracy between this OES-Fed and FedAVG. In other words, the larger the shaded area, the larger the difference. When the dataset was MNIST, the OES-Fed model was used to make more substantial improvement in the accuracy of the worse vehicles. When the dataset was CIFAR-10, the accuracy of each vehicle in the OES-Fed model improved by 30% compared to the FedAVG model. For the vehicle classification dataset, the accuracy of each vehicle in the OES-Fed model improved by 40% compared to the FedAVG model.

## DISCUSSION

As seen in Table 6, the model accuracy of OES-Fed was 2.36% higher than the accuracy of the FedAVG model and 44.46% higher than the accuracy of the FedSGD model in the MNIST. The model loss of OES-Fed was 0.23 lower than the loss of the FedAVG model and 2.11 lower than the loss of the FedSGD model. Also, the AUC value of OES-Fed was 0.22 higher than the AUC value of FedAVG and 0.32 higher than the AUC value of FedSGD. In the CIFAR-10, the model accuracy of OES-Fed was 42.02% higher than the accuracy of the FedAVG model and 47.6% higher than the average accuracy of the FedSGD model. Moreover, the model loss for OES-Fed was 0.38 lower than the average loss for FedAVG and 0.44 lower than the average loss for FedSGD. The AUC value for OES-Fed was 0.14 higher than the AUC value for FedAVG and 0.17 higher than the AUC value for FedSGD. In the vehicle classification dataset, the model accuracy of OES-Fed was 32.4% higher than the accuracy of FedAVG model and 68.0% higher than the average

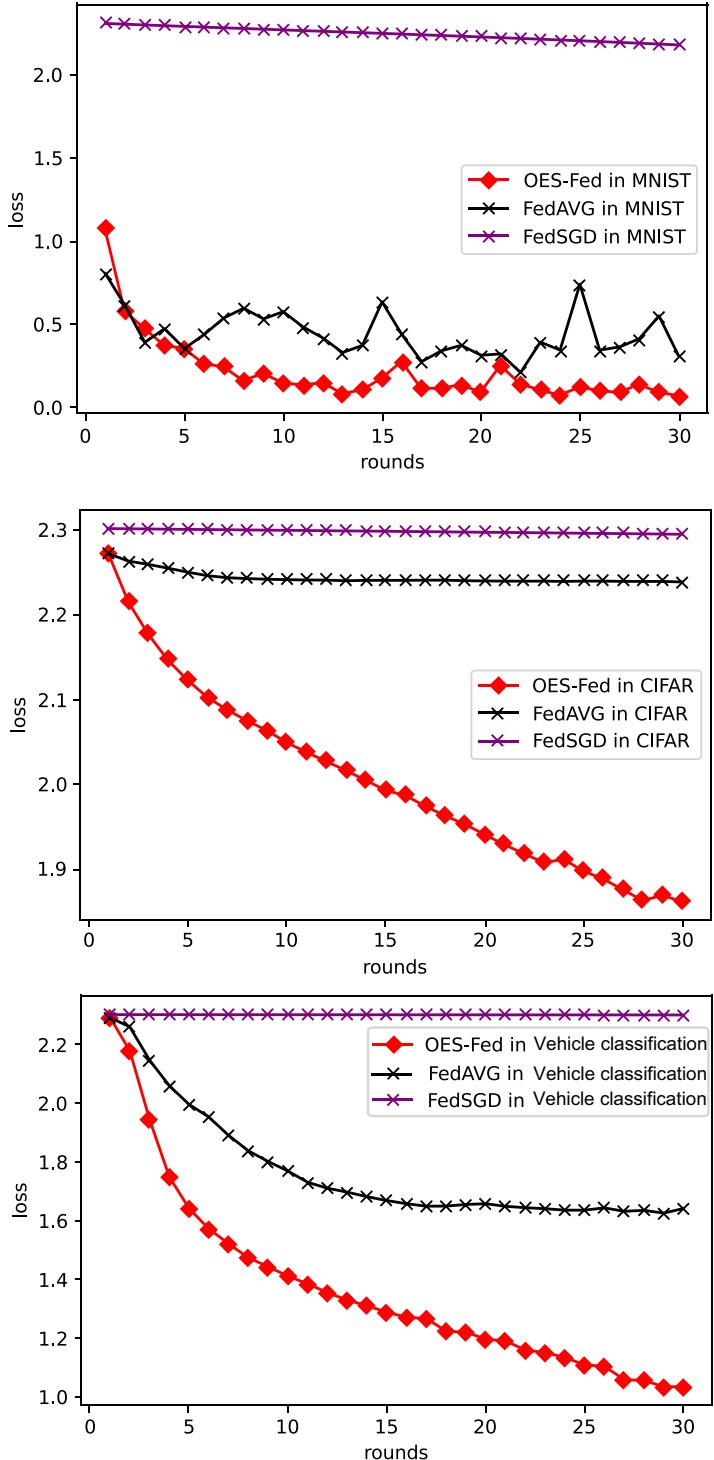

**Figure 6 Comparison of loss values for each model for MNIST, CIFAR-10 and the vehicle classification datasets using different models and non-iid data settings.**

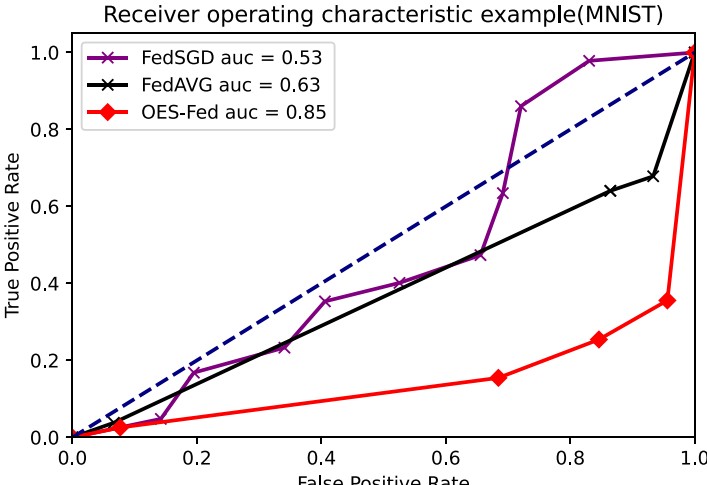

**Figure 7 Comparison of AUC values for each model for MNIST using different models and non-iid data settings.**

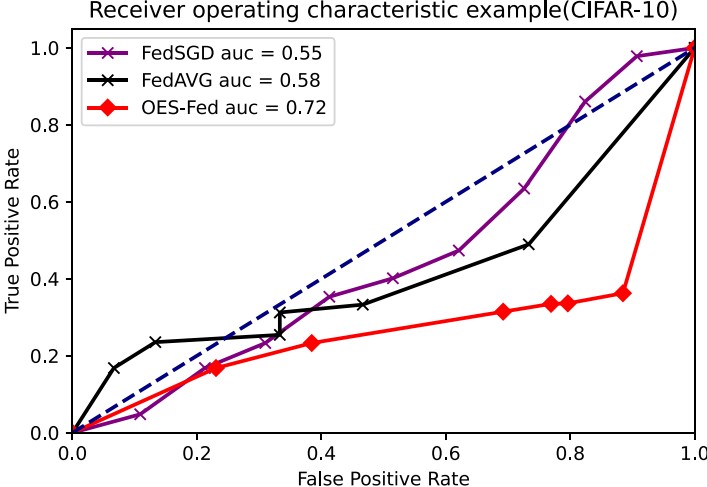

**Figure 8 Comparison of AUC values for each model for CIFAR-10 using different models and non-iid data settings.**

accuracy of FedSGD model. Moreover, the model loss for OES-Fed was 0.59 lower than the average loss for FedAVG and 1.27 lower than the average loss for FedSGD. The AUC value for OES-Fed was 0.27 higher than the AUC value for FedAVG and 0.33 higher than the AUC value for FedSGD.

In summary, the OES-Fed model can better filter out noise data for the global model. It can also greatly improve the recognition accuracy of the model on image data while ensuring communication efficiency. Since only a small number of training clients are removed, the algorithm is still able to maintain the security of the private data of the IoV in the federated learning framework.

Receiver operating characteristic example(the vehicle classification)

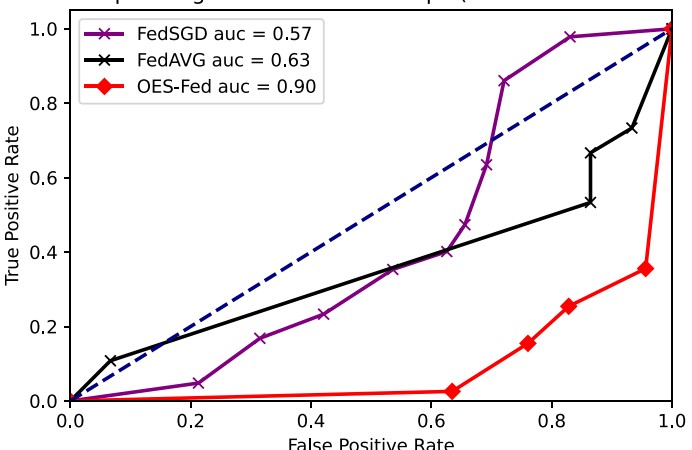

**Figure 9 Comparison of AUC values for each model for the vehicle classification dataset using different models and non-iid data settings.**

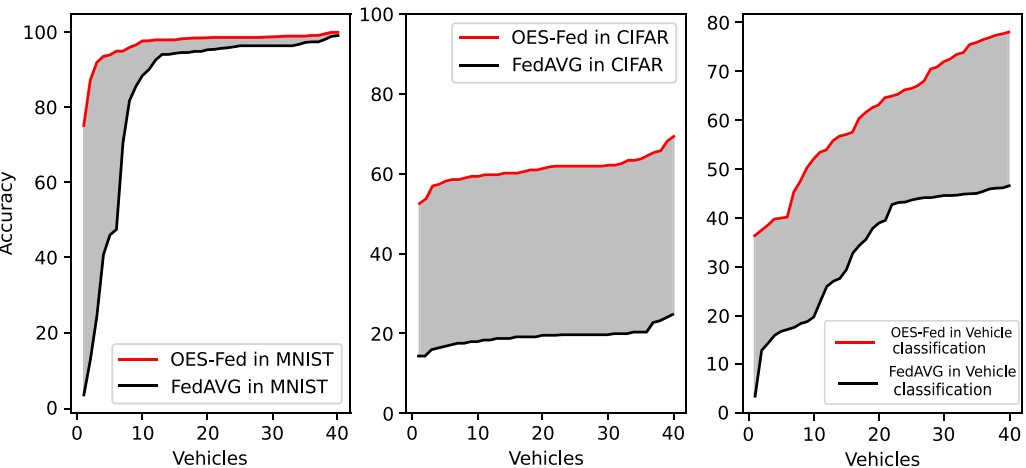

**Figure 10 Accuracy comparison of all clients in the last round for the MNIST dataset, CIFAR-10 dataset and the vehicle classification dataset using the OES-Fed model and FedAVG model.**

**Table 6 Comparison of accuracy, loss values and AUC values for the MNIST dataset, CIFAR-10 and the vehicle classification dataset using FedAVG, FedSGD and OES-Fed.**

|         |          | MNIST | CIFAR-10 | Vehicle classification |
|---------|----------|-------|----------|------------------------|
| FedSGD  | accuracy | 54.20 | 14.15    | 9.9                    |
|         | loss     | 2.18  | 2.30     | 2.30                   |
|         | AUC      | 0.53  | 0.55     | 0.57                   |
| FedAVG  | accuracy | 96.30 | 19.73    | 45.5                   |
|         | loss     | 0.30  | 2.24     | 1.62                   |
|         | AUC      | 0.63  | 0.58     | 0.63                   |
| OES-Fed | accuracy | 98.66 | 61.75    | 77.9                   |
|         | loss     | 0.07  | 1.86     | 1.03                   |
|         | AUC      | 0.85  | 0.72     | 0.90                   |

## CONCLUSION

In this article, a federated learning framework based on Outlier Detection and Exponential Smoothing (OES-Fed) is proposed for IoV networks. The OES-Fed framework realizes noise data filtering while solving the isolated data island by adopting the basic framework of federated learning. By implementing the current and historical perspective of noise data filtering, the accuracy is higher, the loss is lower and the AUC is better to significantly improve the training efficiency of federated learning. To sum up, the framework in this article can effectively optimize the noise data filtering process of federated learning in the IoV. Our proposed OES-Fed framework can effectively filter noise data, but there are still some deficiencies. How to deal with emergencies in the IoV in time and how to apply the OES-Fed framework in the fields of face recognition or network video security would be our further research focus.

### Funding

This work was supported by the Higher Education Department of the Ministry of Education of P. R. China, Industry and University Cooperation Collaborative Education Project, under Grant 202002118061 and by the young and middle-aged teachers' basic ability improvement of Guangxi colleges in 2022 under Grant 2022KY1296. There was no additional external funding received for this study. The funders had no role in study design, data collection and analysis, decision to publish, or preparation of the manuscript.

### Grant Disclosures

The following grant information was disclosed by the authors:
The Higher Education Dept. of the Ministry of Education of P. R. China.
Industry and University Cooperation Collaborative Education Project: 202002118061.
Guangxi Colleges in 2022: 2022KY1296.

### Competing Interests

The authors declare that they have no competing interests.

### Author Contributions

- Yuan Lei conceived and designed the experiments, performed the experiments, performed the computation work, authored or reviewed drafts of the article, and approved the final draft.
- Shir Li Wang conceived and designed the experiments, performed the experiments, performed the computation work, prepared figures and/or tables, authored or reviewed drafts of the article, and approved the final draft.
- Caiyu Su performed the experiments, analyzed the data, prepared figures and/or tables, authored or reviewed drafts of the article, and approved the final draft.

- Theam Foo Ng performed the experiments, analyzed the data, performed the computation work, authored or reviewed drafts of the article, and approved the final draft.

## Data Availability

The raw data is available in the Supplemental Files.

CIFAR-10 python is available at the University of Toronto:

http://www.cs.toronto.edu/~kriz/cifar.html.

The "vehicle classification dataset" (913 MB) is available at AIStudio and accessible using GitHub:

https://aistudio.baidu.com/aistudio/datasetdetail/95878?lang=en.

## Supplemental Information

Supplemental information for this article can be found online at http://dx.doi.org/10.7717/peerj-cs.1101#supplemental-information.

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
