# Peer review of "OES-Fed: a federated learning framework in vehicular network based on noise data filtering"

_PeerJ Computer Science, doi:10.7717/peerj-cs.1101_

## Round 0.1 · original submission · Major Revisions

You have carried out good work. We have some suggestions to improve the quality of the article. Please make the respective changes from the reviewers and re-submit for another review.

Reviewers 1 & 2 have requested that you cite specific references. You may add them if you believe they are especially relevant. However, I do not expect you to include these citations, and if you do not include them, this will not influence my decision.

·

Basic reporting

Clear and unambiguous, professional English used throughout.

Experimental design

Research question well defined, relevant & meaningful. It is stated how research fills an identified knowledge gap

Validity of the findings

Conclusions are well stated, linked to original research question & limited to supporting results.

Additional comments

1. Authors should increase the figure size for a better understanding of the work.
2. A few images are not clear. It is recommended to identify the unclear images and revise them accordingly.
3. Literature is not covered with recent works that were published in the year of 2021 and 2022.
4. References are limited, better to extend with the publication of 2021 and 2022.
5. Please find the following research works, there is any relavence use it for your literature.
A. MTCEE-LLN: Multilayer Threshold Cluster-Based Energy-Efficient Low-Power and Lossy Networks for Industrial Internet of Things.
B. An Energy-Efficient Routing Scheduling Based on Fuzzy Ranking Scheme for Internet of Things.
C. ETH‐LEACH: An energy enhanced threshold routing protocol for WSNs
D. Arior: Adaptive ranking based improved opportunistic routing in wireless sensor networks
E. Energy-efficient blockchain implementation for cognitive wireless communication networks (CWCNs)
F. Performance analysis of energy efficient opportunistic routing protocols in wireless sensor network

Reviewer 2 ·

Basic reporting

no comment

Experimental design

no comment

Validity of the findings

no comment

Additional comments

1. The English language should be improved to ensure that an international audience can clearly understand your text. e.g., "The OES-Fed framework we proposed can help the IoV better ûlter noise data, and provide a reference for the signiûcant starting ûeld of federated learning in IoV." should repharse. Authors should check the complete paper for same types of corrections.
2. Introduction section may have a figure that explain the working behavious of research area.
3. Literature review and problem formulation are very well explained.
4. conclusion section should include the future aspects.
5. To strengthen the introduction part , you can refer the following articles
a). Singh, Gagan Deep, Manish Prateek, Sunil Kumar, Madhushi Verma, Dilbag Singh, and Heung-No Lee. "Hybrid genetic firefly algorithm-based routing protocol for VANETs." IEEE Access 10 (2022): 9142-9151.
b). Singh, Gagan Deep, Sunil Kumar, Hammam Alshazly, Sahar Ahmed Idris, Madhushi Verma, and Samih M. Mostafa. "A novel routing protocol for realistic traffic network scenarios in VANET." Wireless Communications and Mobile Computing 2021 (2021).
c). Chithaluru, Premkumar, Sunil Kumar, Aman Singh, Abderrahim Benslimane, and Sunil Kumar Jangir. "An Energy-Efficient Routing Scheduling Based on Fuzzy Ranking Scheme for Internet of Things." IEEE Internet of Things Journal 9, no. 10 (2021): 7251-7260.

Reviewer 3 ·

Basic reporting

This paper presents a federated learning framework in vehicular network based on noise data filtering. This paper is well written, and the content is very substantial. However, there are some problems with the manuscript,

Experimental design

The main application scenario of this paper is IOV, but the data set used in the experiment is still traditional, such as MNIST dataset and CIFAR-10 dataset. In my opinion, some data sets related to IOV can be collected for experiments to verify the effect of this model in specific application scenarios

Validity of the findings

The idea proposed in this paper is very novel, but according to the existing conditions, whether the local training model of vehicle end can be realized.

Additional comments

1.The content of this paper is well written, but there are some symbolic problems, for example, in section 1.2, there are symbolic errors in the fifth line of the third paragraph. Wn should be written as w_n. Please check the spelling and symbols in the paper carefully.
2.In formula 5 in the paper, I think there is something wrong with your formula. According to the formula in the paper, I think the correct formula should be minf(w)≜1/N ∑_(n=1)^N▒〖f_n (w) 〗, where f_n (w) is the loss function of the local model. Please check the formula writing in the full paper.
3.In formula 26 in the paper, you define the loss value used for an experiment. However, I can’t understand the meaning of loss value clearly. Please explain the specific meaning of the loss value in the paper.
4.There are some works for possible improvement in content and structure: ‘Entropy-based redundancy analysis and information screening’, ‘MSTA-Net: Forgery Detection by Generating Manipulation Trace Based on Multi-scale Self-texture Attention’, ‘MTD-Net: Learning to Detect Deepfakes Images by Multi-Scale Texture Difference’.

---

## Round 0.2 · Major Revisions

Good work, we have a few suggestions but your article is effective and of quality. Please complete the suggestions from the reviewer. Please complete and resubmit.

Reviewer 2 ·

Basic reporting

all review comments are well addressed

Experimental design

all review comments are well addressed

Validity of the findings

all review comments are well addressed

Reviewer 3 ·

Basic reporting

Some sentences in the paper are long, which leads to certain obstacles to the reading of the paper. You can modify the sentence structure,such as the sentence on line 271.

Experimental design

In Figure 4 of the experimental part, the red part in the middle and the red part on the right are completely consistent with the experimental results. Is it a coincidence? It is recommended to increase the number of experimental rounds to verify the experimental results.

Validity of the findings

None

Additional comments

The author has replied to some questions, but there are still some questions that have not been completely revised. I have found some new problems, and I hope the author will make serious revisions.

In the last review, I introduced the author to some works that could improve the content and structure of this paper, but the author did not fully refer to them. Although some of these works doesn't quite match this paper, I think their structure and language can help the authors, so I hope the authors will consider referencing them and citing them.

---

## Round 0.3 · accepted · Accept

Please carry on the good work.

Reviewer 3 ·

Basic reporting

None

Experimental design

None

Validity of the findings

None

Additional comments

None